# Psychological Determinants in Biathlon Performance: A U23 National Team Case Study

**DOI:** 10.3390/sports12020038

**Published:** 2024-01-23

**Authors:** Frank Eirik Abrahamsen, Andreas Kvam, Stig Arve Sæther

**Affiliations:** 1Department of Sport and Social Sciences, Norwegian School of Sport Sciences, 0806 Oslo, Norway; 2Department of Sociology and Political Science, Norwegian University of Science and Technology (NTNU), 7491 Trondheim, Norway; andrkvam@gmail.com (A.K.); stigarve@ntnu.no (S.A.S.)

**Keywords:** self-efficacy theory, mental training, coping, sports, biathlon

## Abstract

Background: The present investigation examined what psychological factors athletes perceived to impact their competition performance and what training strategies and focus the athletes considered to be the most important. Methods: We recruited six participants (three females, three males) from the Norwegian Biathlon Federation’s national U23 and junior teams, and all participated. We used semi-structured interviews to gather the data and used thematic analyses to examine our findings. Results: The findings centered around the intricate relationship between psychological factors, particularly self-efficacy, anxiety, attention control, and performance, in biathlon shooting. Conclusions: Implementing a holistic approach to biathlon training entails harmonizing physical and psychological elements with personalized psychological training regimens.

## 1. Introduction

Biathlons are a winter sport that combines cross-country skiing and rifle shooting, requiring a unique mix of endurance, skiing skill, and marksmanship. The first Biathlon World Championship was held in Austria in 1958, and the sport joined the Olympic Games (OG) in 1960 (men) and in 1992 (women). In the 2022 OG, more than 200 athletes competed from more than 30 countries in 11 events.

Biathlon is a complex sport that combines the technical aspects of cross-country skiing with point shooting, requiring high levels of precision, extreme focus, technical ability, and small margins [1,2]. The performance demands of endurance and shooting require the athletes to maintain high heart rates and deal with competition and time pressures, making attentional focus and arousal regulation crucial during races. The following quote from the Olympian biathlete Clare Egan illustrates the challenges in a chronicle [3]: “The physical things are difficult—using all your muscles and pumping your heart as fast as you can… But the mental piece is the biggest challenge of biathlon”.

Despite the demanding requirements of endurance and attentional capacity in biathlons, research on this sport remains surprisingly scarce. Laaksonen and colleagues [1] reported only 79 studies on biathlons, while cross-country skiing yielded ten times more results. In their review, Laaksonen and associates [1] recognized critical performance elements such as rifle stability, shooting position, triggering, and cardiac cycle. Biathlons are primarily considered an endurance sport, based on the cross-country skiing element. Even though psychophysiology has a long tradition in shooting, shooting in biathlons is under-researched [1]. The same year, Laaksonen and colleagues [2] advocated that elite performers undergo training in conditions resembling competitions. This approach not only enhances accuracy but also minimizes time loss during shooting.

The authors above suggest a training method akin to what is commonly referred to as simulation training in sports psychology [4]. “In the main, the research that has been published highlights the efficacy of various perceptual-cognitive training interventions” [4] (p. 241). For biathletes, it is crucial to maintain focus while shooting under high-pressure situations amidst various stimuli such as wind, opponents, speakers, and spectators, which can interfere with their performance. Specific distracting thoughts, such as being preoccupied with the fear of missing a shot or obsessively fixating on the outcome, can also have a detrimental impact. Replicating some distractors in training is viable, whereas others, such as the crowd’s impact, can be easily experienced during competitions. In addition to simulation training, mental skills training is essential to prepare for competitions, including various techniques like imagery, self-talk, goal-setting, and physical relaxation.

### 1.1. Self-Efficacy Theory

An influential theory that provides insights into how athletes can effectively handle race demands is the self-efficacy theory (SET), developed by Bandura [5] in 1977. Initially proposed as a stress theory, SET has evolved into a motivational theory, which explains how self-efficacy impacts beliefs, effort, and performance. Self-efficacy refers to an individual’s confidence in executing the necessary behaviors to achieve a desired outcome [5,6,7]. Although the skills are there, Clare Egan illustrates how difficult it might be: “I have this task I’ve done thousands of times that I’m trying to repeat. I know that I’m going to have distractions. The person next to me hit all of the targets. The fans are screaming. The person on the loudspeaker says, “Here’s Clare Egan from the U.S.A. Let’s see if she can hold it together” [3].

Bandura identifies four primary sources of efficacy information, the first being former mastery experiences of a task. According to Bandura [5,7], this source is the most significant and authentic evidence of an individual’s ability to meet task demands successfully. Past successes are likely to enhance self-efficacy beliefs, while past failures can undermine them. The second source of self-efficacy beliefs in Bandura’s theory [5], termed vicarious experiences, including symbolic modeling—analogous to imagery. The third source was termed verbal persuasion, including self-talk, and the fourth and final was coined emotional arousal [5]. Bandura argued that the efficacy sources are receding in order, with past mastery experiences considered to be the most important.

### 1.2. Self-Efficacy, Performance, and Mental Training

Self-efficacy theory has been a popular framework for studying sports in the past. Several reviews have been published [8,9,10,11,12,13]. These reviews generally consider a positive relationship between self-efficacy and performance [8,9]. Similarly, self-confidence and performance reviews report comparable results [10,11,14]. Athletes with high self-efficacy beliefs may better overcome challenges and achieve goals, see [8,12]. Feltz and colleagues [8] discuss the reciprocal relationship between self-efficacy and performance. Although past performances affect self-efficacy, the relationship between self-efficacy and performance cannot be disregarded according to these researchers. In their annotated bibliography, short abstracts from every study conducted to that date (2001) were included, and most of the included studies emphasized summer sports and activities. In their review of self-regulation (including self-efficacy), the authors conceded: “Despite the value of researching the role of self-efficacy in endurance performance, we observed that research examining self-efficacy and performance in experienced athletes is scarce” [15] (p. 250). The authors continue that effective coping is crucial for endurance performance, making it necessary to explore the connection between coping self-efficacy and endurance. Given a lack of self-efficacy research both on winter sports and elite sports, our focus on biathlons aims to make a small contribution in this regard.

Biathlons are primarily a winter sport, although there are a few roller-skiing competitions in the summer. While there are technical differences between skiing and roller skiing, the pressure of competition, shooting demands, and cardiovascular strain can be comparable. Recognizing that winter competitions hold more prestige than summer races, winter games may intensify the competition pressure. Concerning SET, biathlon athletes thus have fewer opportunities to encounter competition pressure and master challenging experiences in specific season periods.

Consequently, the remaining three sources of efficacy beliefs become crucial for building athletes’ confidence and belief in their abilities. Utilizing mental training techniques like imagery and self-talk is recommended [15,16], sharing similarities with vicarious experiences (i.e., imaginary experiences) and verbal persuasion (i.e., self-talk), respectively, see [5]. Due to the efficacy sources utilized in SET and the similarities with various mental skills and training techniques, SET undergirds the theoretical foundation for the present study.

In their review, Lochbaum and colleagues [13] delved deeper into potential mechanisms underlining the confidence–performance relationship. They reported that closed-skills sports, individual sports, and sports with competitions shorter than 10 min had a stronger relationship between confidence and performance than open-skill sports, team sports, and sports over 10 min. One might consider cross-country skiing and shooting as open-skill, sports since the athlete must adapt to the environment. However, are biathlons a sport comprising two sports that, in many ways, are incompatible? One sport is an endurance sport (over 10 min), and one is a fine motor sport (under 10 min). Evans’s sports psychologist described preparation for such conditions [3]. Sean McCann (USOC) helps Egan and other biathletes develop breathing techniques to calm themselves during the brief transition from racing to shooting. Mindful breathing slows their heart rate just enough to improve their accuracy in the target portion of the competition [3]. The anecdotal report of Egan and McCann signifies the importance of mental skills and mental training for biathletes.

### 1.3. Study Focus

However, which psychological factors and mental training strategies biathletes consider essential is mainly undetermined. More research is needed, especially considering that the sport has two distinctive components demanding different skills. Thus, the present investigation examined what psychological factors athletes perceived to impact their competition performance and what training strategies and focus the athletes considered to be the most important.

## 2. Materials and Methods

The current study embraces the social constructivism paradigm [17,18], emphasizing individual-specific worldviews shaped by interactions with cultural norms, values, and the meanings individuals assign to these interactions [19]. Thus, we embraced an explorative and inductive qualitative approach, seeking to understand the athletes’ knowledge of their own performance reality. As recommended to gain insight into the athlete’s perspectives [20], this study employed a qualitative and exploratory integrative case study design, utilizing semi-structured interviews. This study was conducted in line with the Declaration of Helsinki and approved by Norwegian Social Sciences Data Services (reference no. 802411).

### 2.1. Participants

We recruited six participants (three females, three males) from the Norwegian Biathlon Federation’s national U23 and junior teams, and all participated (see Table 1). To be eligible for this study, the participants must have been a part of the national junior team in the previous competition season, with an average age of 20.8 (SD 0.6) and an average of 12.5 years of experience. Furthermore, all the athletes had previously been part of specialized school sports programs between the ages of 16 and 19, and several had won national and international junior championship medals in the prior two seasons.

### 2.2. Interview Guide

An interview guide was developed based on a review of topics related to performance development in sports, biathlon shooting determinants, and performance development in biathlons. The guide included warm-up questions to gather information about the participants’ sports backgrounds and their introduction to biathlons, see [21,22]. The guide also contained open-ended questions on critical topics such as essential types of shooting training, typical shooting exercises, and reflections on shooting training during their junior years. We provide example questions in Table 2. The goal of using a semi-structured interview technique was to allow the participants to express their perspectives freely. A pilot test with another junior biathlete ensured the quality of the interview guide, which resulted in modifications to the question sequence to connect related topics. According to Moser and Korstjens [22], piloting the interview guide helps assess its relevance and content analysis and identifies any need for reformulation. The pilot testing uncovered that the interview exceeded the expected duration, but this extended time frame proved instrumental in uncovering valuable insights for the explored themes and topics.

### 2.3. Data Collection

We completed six semi-structured interviews to gain insight into the athletes’ perceptions of vital psychological determinants, coping strategies, and the development of mental skills in biathlons. The U23 national teams have only twelve athletes (six male and six female); thus, the six participants in this study encompass a highly elite group. The interview guide helped us inquire about the factors above and allowed for relevant follow-up questions. The participants freely elaborated on the topics in question, following recommendations made by Moser and Korstjens [22,23].

Due to the athletes being in-season and needing flexibility concerning their travel schedules and the risks associated with contracting COVID-19 during a competition period, we completed the interviews individually and digitally using either Microsoft Teams or Zoom. Within one month, the interviews concluded, lasting 45 min to 1 h each. The interviewer fully transcribed each interview, totaling 54 pages of transcribed text (Arial 12, line spacing 1.5).

### 2.4. Data Analysis

The interviews were recorded using an audio recorder and then fully transcribed. Unique identification codes (I1 to I6) were assigned to each participant to ensure participant anonymity. The following steps, as suggested by Braun et al. [24], were taken to analyze the data: (1) transcribing the interviews, (2) reading and re-reading the data to generate initial codes (e.g., self-efficacy and attention focus), (3) using deductive codes and identifying lower-order themes under the initial codes, (4) laying out the main topics from the material, (5) reviewing the final categories and under-themes, and (6) writing the final report and presenting the data.

After conducting the six steps outlined above, we reviewed the data and identified three central topics. We present these topics in detail in the results and discussion section: (a) self-efficacy, (b) attentional focus and arousal regulation, and (c) mental training/strategies for coping and development. Finally, all the authors discussed and agreed on the different themes.

### 2.5. Research Rigor

To ensure meaningful coherence between the purpose of the study and the procedures that were followed, we tried to be as transparent as possible about the processes that lead us to our analytical findings by attempting to continually verify our analysis and our critical interpretations of the data. This enabled us to test both our procedures and our data analysis. It furthermore enabled us to make explicit our pre-conceptions, our sensitivities to the research contexts in which we were working, and to establish an analytical distance that would allow us to better reflect on our interpretations [17]. During the data collection processes, we also discussed our results and our own interpretations to ensure peer agreement and validity. The transparency of the analysis and judgement processes add to the research credibility established by carrying out multiple interviews to allow for participant reflections and having additional researchers act as critical friends [17,18].

## 3. Results and Discussion

Reiterating, the present study examined what psychological factors the athletes perceived to impact their competition performance and what training strategies and focus the athletes considered to be the most important. This study encompasses three primary themes: (Section 3.1) *self-efficacy*, (Section 3.2) *attentional focus and arousal regulation*, and (Section 3.3) *mental training and strategies for coping and development*. First, we will introduce each theme relating to SET and previous research. Then, we will complete the discussion with general thoughts and practical implications (Section 3.4) before ending the chapter with limitations and future research possibilities.

### 3.1. Self-Efficacy

Self-efficacy and confidence emerge as central factors for shooting performance that were consistently emphasized by all informants as crucial contributors to performance and skill development in shooting, if not the most momentous—corresponding to Moritz and colleagues’ review [9]. Three different athletes point this out in the following quotes:

I3:
*If you do not believe the things you do are the right thing to do, I think it is challenging to get quality. To have this belief is essential to me.*


I5:
*Everyone at this level knows how to shoot. When you are at a certain level, it can be challenging to point out why it is working or, on the other side, not working. I think this is based on self-efficacy and the belief in your abilities.*


I2:
*The psychological elements are so crucial in competitions. If you are just a bit unsure of your performance, you might quickly shoot two misses.*


Two athletes point to psychological skills as the essential factor—the foundation for their success on the biathlon track. When reflecting upon their perspectives, one can find I3’s words to be insightful, as the quote resonates with competitors’ challenges in striving for excellence in all demanding sports. In the realm of biathlons, where split-second decisions and precision can determine victory or defeat, the power of self-belief emerges as a focal force. I5’s observation that everyone knows how to shoot sheds light on the fact that technical prowess alone cannot hit the mark consistently. The difference lies in athletes’ faith in their abilities—the essence of self-efficacy, as Bandura outlined [5]. I2’s words echo how fascinating it is to observe how a hint of uncertainty can reverberate downrange, resulting in missed shots. The athlete’s uncertainty accentuates the delicate interplay between mental fortitude and physical execution in biathlons. When the mind wavers, so too does the aim.

The athletes’ collective insights converge on a common theme—a profound recognition that self-efficacy is the linchpin in their performance. This sentiment gains empirical support from the works of Feltz [8], Kočergina [25], and Ortega and Wang [16], whose inquiries accentuate the transformative influence of self-efficacy and confidence in athletic excellence. As the biathletes assert, their shooting prowess results from mechanical repetition, a fusion of psychological conditioning, technical mastery, and an unyielding belief in their capabilities. Their descriptions of needing to attain a 100% alignment in shooting position vividly illustrate how psychological factors intertwine with the physical.

#### 3.1.1. Budding Self-Efficacy

I3:
*I think I would say psychological skills are the most important to manage the shooting. Of course, it is essential with a relaxed shooting position and rifle stability, but for me, the psychological skills are what I think are the most important.*


I4:
*I look at [being] sure of oneself and self-efficacy as the most important determinants.*


Within the competitive sports milieu, wherein precision and psychological resilience are pivotal, the salience of psychological skills becomes resoundingly apparent. Articulated by I3’s assertion that psychological skills are important in the shooting part, an exploration into the intricacies of marksmanship reveals that while tangible aspects such as a relaxed shooting posture and firearm stability are undeniably germane, the cornerstone of accomplishment rests upon the psychological underpinning, see also [26,27]. These sentiments resonate with I4’s perspective, which ascertains that self-assurance and self-efficacy are foundational in determining shooting performance.

This conceptual pivot unveils a discernible dichotomy, encapsulating two overarching psychological themes threading through their discourses. The first theme, self-efficacy, illuminates the prominence of cultivating faith in one’s competencies,—internalizing the conviction that success is attainable via persistent endeavor and skillful execution. The second theme pertains to attentional focus and regulating arousal states, uncovering the workings of maintaining equanimity amid intensified competition. The art of channeling cognitive focus with precision and calibrating arousal levels commensurate with task demands assumes the guise of a psychological tightrope walk, for excellent discussions, see [27,28]. Conversely, attentional focus and arousal regulation may bolster self-efficacy, affording athletes the cognitive tools to internalize their latent capacities and flourish under duress.

#### 3.1.2. Self-Efficacy Development

The insights in the quotes below show a profound interplay between technical proficiency, self-efficacy, and targeted training strategies that athletes employ to optimize their performance, particularly in the critical period preceding competitions.

I2:
*After considerable time in my standing shooting position, I often feel confident and strongly believe in my performance.*


I2’s reflection unveils a titillating phenomenon: the gradual cultivation of confidence through sustained immersion in the standing shooting position. This process of immersion engenders a transformative conviction in one’s performance. This insight underlines the symbiotic relationship between physical practice and psychological conviction. In Norway, biathlon athletes only start standing shooting from the age of 15–16, see [29], which is later than most competitors abroad. While this has positive safety implications, it might also give less time to develop this skill for senior-level competitors. The following quotes show the importance of training over time:

I6:
*It is so important to me because of the specificity and coping—this type of training boosts my self-efficacy and confidence.*


I3:
*To get many repetitions and implementations of what I am supposed to do in a competition is vital to me. The specificity is an integral part of my shooting philosophy.*


The collective perspective articulated in the quotes underscores the critical role of technical competence as the bedrock for self-efficacy. Athletes, as attested by the informants, derive assurance and empowerment from aligning the diverse technical facets, fortifying their self-efficacy. Their testimonies emphasize that self-efficacy is not solely a product of abstract belief, but an outcome derived from the tangible fusion of technical precision and mental preparedness. As evident in I6’s quote, coping and specificity in training bolster self-efficacy and confidence. This account also emphasizes the significance of simulating the competitive context, fostering a mental congruence between training and actual events. I3’s perspective also highlights the essence of repetitions and implementations. Such deliberate immersion aligns with the principles of experiential learning and the biathlon federations’ development plan (see above). When athletes acclimate to pressure scenarios, this may reinforce self-efficacy and bolster conviction in their abilities. A holistic approach, anchored in intertwining physical and cognitive preparations, is a testament to the multifaceted nature of prowess in biathlon performance. Corresponding to our findings, Blumenstein and colleagues [30] argue that athlete training emphasizes three key phases—preparatory, competition, and transition—where all phases comprise physical, technical, tactical, and psychological readiness. For optimal proficiency, these aspects should harmonize within each phase. Their article was about judo; however, the seamless integration of sports psychology with the other segments must align with phase-specific goals and connect with physical, technical, and tactical readiness.

### 3.2. Attentional Focus and Arousal Regulation

In line with previous research [16,26,31,32], attention and arousal regulation were considered to impact the performance of the athletes in this study. Strategies to cope with this and enhance one’s attention and focus are crucial. One of the informants talks about his experiences in this way:

I3:
*I think it can be important to tell someone, or just yourself, about your plan for the competition or session. I used to have a chat with my coach before competitions, where I talked about my well-defined plan. I have experienced pressure in competitions where my attention floats away, and I cannot switch back to my tasks. But when I talk about it out loud in advance, I feel a more robust control, and it is easier to switch back to the tasks I told my coach about.*


I3’s insights provide a valuable perspective on the role of verbalization and pre-competition planning in optimizing performance within the realm of sports. Articulating one’s plan, either to oneself or to an external party like a coach, may serve as a mechanism for enhancing cognitive control, maintaining focus, and mitigating performance pressure. Communicating one’s plan—internally or in dialogue with a coach—may reinforce mental preparedness. By externalizing thoughts, athletes provide structure and tangibility to their strategies. Furthermore, discussing a well-defined plan with a coach before a competition can be perceived as a form of commitment. The commitment that follows telling the plan to others may function as goal setting, an established fundamental mental technique [33].

Also, I3’s observation about experiencing attentional fluctuations in competitions accentuates the vulnerability of concentration to external pressures. Different theories of how stress may affect focus and concentration include turning towards and turning away, see, for instance, [34,35]. As Gray outlines, the proponents of the “turning toward” perspective contend that choking stems from the redirection of attention toward skill execution, leading to subsequent performance disruptions [34] (p. 595). At the same time, adherents of the “turning away” viewpoint posit that choking arises from attention being drawn away from skill execution, resulting in a diminished focus on task-relevant information. I3 appears to have honed a skill that enables the recalibration of focus as required in competitive events. While the precise nature of this focus realignment—whether directed inwardly or outwardly—remains unclear in the quote, I3’s insights have facilitated the formulation of effective attention management techniques. These insights highlight that contemplating skill execution and performance could prove beneficial at this advanced stage of competition.

#### 3.2.1. Process Goals in the Competitive Situation

The informants below (I1, I3, I6) emphasize the relevance of having few tasks to work with simultaneously. The collective insights conveyed through these quotes reveal a nuanced interplay between confidence, attention management, task focus, and performance optimization within a competitive context. Each athlete’s perspective underscores the profound effect of task quantity, focus clarity, and self-assurance on their ability to navigate high-pressure situations effectively.

I1:
*The fact that I am so confident in the one or two tasks I have prepared makes me feel I have more capacity for sharp attention.*


I3:
*If I have too many tasks, I have experienced that my focus floats away. I get eager and do things too fast. When I leave the range, I ask myself, “What happened there?” It is just flashing by.*


I6:
*I used to have only one task and do that task 100%.*


I1’s observation points to the symbiotic relationship between confidence and attentional acuity. The conviction in one’s prepared tasks may generate a mental surplus—a reservoir of heightened attentiveness, potentially safeguarding against distractions. This sentiment again echoes that assurance in one’s capabilities breeds the mental requirements for optimal performance, see [9]. I3’s insights provide a distinction, highlighting the potential pitfalls of task overload. One athlete’s experience of attentional drift and the sensation of tasks flashing by underlines the challenge of juggling multiple demands, especially simultaneously.

The phenomenon of becoming overeager and hastily executing tasks underscores the potential for a fragmented focus. I3’s candid reflection after leaving the range—questioning what transpired—epitomizes the dissonance between intention and execution when confronted with an excessive cognitive load. This reflection underscores the delicate balance between task complexity and focused execution, emphasizing the need for a strategic approach to attention allocation in elite sports. I6’s viewpoint revolves around singular-task dedication, suggesting that immersing in one task may enhance an athlete’s cognitive precision. However, how this approach would adapt to sudden environmental changes like wind shifts at the shooting range remains unclear. While I6’s method could simplify situational demands, reducing the risk of scattered attention or hastened tasks, it might falter when vital situational cues go unaddressed. Notably, the first author’s extensive applied work in biathlons reveals athletes’ tendencies to shift focus progressively: from wind direction upon entering the shooting range to posture and trigger-pull. This adaptive strategy resonates with I6’s perspective, indicating the focus-shifting context that likely underpins I6’s statement. Ultimately, while singular-task immersion offers benefits, its adaptability in dynamic scenarios necessitates considering situational complexities.

#### 3.2.2. Coping Is about Tolerating Anxiety

The informants underscore an intricate interplay involving self-efficacy, traits of anxiety, arousal, and tension. Extended intervals devoid of a sense of mastery over core shooting proficiencies have, on occasion, induced subdued manifestations of anxiety among the informants. Bandura’s self-efficacy theory asserts that an individual’s belief in their capability to execute specific tasks influences their actions. The concept of generalizability [5] extends this concept, suggesting that successful performance in one domain can bolster confidence in other areas and situations, fostering a broader sense of competence. As Bandura outlines, e.g., [5,7], negative performance accomplishments might undermine their efficacy beliefs.

I5:
*I have experienced phobic anxiety about entering the shooting range during competitions. After many bad experiences over a long period, I felt anxiety and increased physiological arousal coming into the shooting range.*


I4:
*My experience is that low self-esteem and self-efficacy make it easier to choke under pressure […] In periods with lower self-efficacy, it is harder to control physiological tension and arousal, and I feel less psychological pressure can make me choke.*


I5’s account illuminates the impact of anxiety related to entering the shooting range during competitions. The accumulation of adverse experiences over an extended period engenders a sense of anxiety and heightened physiological arousal upon approaching the shooting range. This phenomenon underscores the depth of emotional conditioning, where past negative encounters become powerful triggers, influencing an athlete’s psychological and physiological states—in this case, negatively. I4’s perspective delves into the nexus between self-esteem, self-efficacy, and choking under pressure, where the connection between low self-esteem, self-efficacy, and susceptibility to choking becomes evident. Furthermore, controlling physiological tension and arousal becomes challenging for I4 in periods marked by diminished self-efficacy. The informants highlight various negative experiences in competitions that led to anxiety, heightened physiological arousal, and tension, thereby exacerbating the challenges of the shooting situation. These findings align with previous research, which indicates that uncontrolled increases in arousal and tension are associated with decreased performance [36,37,38]. Gray argues that the “turning away” perspective explains pressure’s impact on performance during action planning and post-movement reflection [34] (p. 606). In contrast, the “turning toward” viewpoint clarifies pressure’s influence during movement execution. Gray raises inquiries about the interplay of arousal and attention control, highlighting the necessity for further research in sports at large and, we would argue, particularly in the context of biathlons [34] (p. 606).

### 3.3. Mental Training and Strategies for Coping and Development

Given the informants’ emphasis on the significance of psychological skills, particularly attentional focus, exploring how these factors influenced their training characteristics became intriguing. When asking the informants about their methods for cultivating psychological skills and enhancing performance, for instance, informant number 5 said this:

I5:
*I think it is crucial to prioritize psychological training, but it is so abstract that it makes it difficult to know what it is and implies. I do not have a structured training plan for developing psychological skills, but I think it is essential to train your head.*


Despite its abstract nature, I5 highlights the importance of prioritizing psychological training in sports. The challenge lies in understanding its implications. Despite lacking a structured plan, I5 emphasizes the necessity of cultivating mental resilience through psychological training. Reasonably, another informant describes the same perception:

I2:
*We do some shooting-specific psychological training with duels, relays, and exercises where you get to feel the pressure and have to focus on your attention. The coach may try to get into our heads by commentating on the shooting underway. However, I rarely have any purely psychological training with the aim of developing skills.*


I2’s account resonates with I5’s sentiment, explicating the training approaches utilized in biathlons. The training activities mentioned—duels, relays, and exercises—imply an attempt to simulate pressure situations and enhance attentional focus. However, the absence of dedicated psychological training for skill development becomes apparent. This observation aligns with I5’s point about the dearth of structured psychological training plans. These two quotes collectively signify a shared awareness among athletes regarding the significance of psychological skills, such as attention focus and attention regulation, for optimal biathlon shooting performance. However, they also underscore a potential gap in their training regimens: while the importance is recognized, a structured approach for nurturing these skills might be missing. This gap indicates a need for more explicit guidelines and frameworks that address the abstract nature of psychological training, enabling athletes to systematically enhance their mental prowess alongside their physical skills.

### 3.4. General Discussion

The present investigation examined what psychological factors biathlon athletes perceived to impact their competition performance and what training strategies and focus they considered to be the most important. The findings centered around the intricate relationship between psychological factors, particularly self-efficacy, anxiety, attention control, and performance in biathlon shooting. The athletes’ insights, as portrayed through various quotes, highlight the critical interplay of these elements.

The athletes’ accounts underline the pivotal role of self-efficacy in shaping performance outcomes through different avenues. A strong belief in one’s capabilities emerges as a foundational determinant of success, mitigating the risk of choking under pressure. This concept aligns well with Bandura’s theory [5], which, in 1977, focused initially on stress. Attention control also emerges as a critical aspect, with athletes strategically shifting their focus at different phases of competition, underscoring its dynamic nature, see [27,28,34]. Anxiety is a complex adversary, obviously having the potential to undermine performance, for discussions, see [34,35,36]. Athletes recount instances of anxiety linked to shooting range entry, highlighting the profound emotional conditioning that can influence psychological and physiological states before the performance task begins. The challenges posed by abstract psychological training come to the forefront. While recognizing its importance, athletes find it challenging to comprehend its implications and integrate it into their daily training routines. Therefore, several researchers consider this to be a critical facet is developing a holistic training approach, e.g., [30], addressing the physical and psychological dimensions of biathlon shooting, also see [1,2]. More explicit frameworks for psychological training, tailored to athletes’ needs, are essential to unlock their potential for mental resilience. This synthesis suggests that, while technical skills remain crucial, enhancing psychological attributes, bolstering self-efficacy, managing anxiety, and refining attention control are integral components in achieving peak performance in the demanding sport of biathlon shooting. Although not examined in the present investigation, implementing a holistic approach to biathlon training entails harmonizing physical and psychological elements with personalized psychological training regimens, as Blumenstein and colleagues [30] recommend with judokas. Combining mental aspects with technical skill refinement may equip athletes to navigate the multifaceted demands of biathlons with confidence and peak performance [37,38].

While this research primarily focuses on youth elite athletes, it is essential to note that the biathletes interviewed are integral members of larger teams. We perceive them as representatives of their respective national teams and expect that the insights gained from this study have relevance for athletes operating at the sub-Olympic level. However, it is essential to recognize that the findings may not be universally applicable across different levels and sports. Nonetheless, we hope the findings provide valuable insights into sports preparation, particularly in a discipline like biathlons, where Norway has consistently achieved Olympic success. In order to improve this research in the future, longitudinal studies might be viable. Such efforts have recently been successfully undertaken with a prospective cohort study on injury and illness in biathlons [39]. To advance our comprehension and augment the sophistication of our professional endeavors within the realm of sports psychology, we aspire to systematically broaden the repository of our cognitive resources and research methods in ensuing scholarly endeavors on biathlons.

## Figures and Tables

**Table 1 sports-12-00038-t001:** Description of the participants.

Informant ID	Gender	Age	Years of Biathlon Experience
I1	Male	19–21	13–15
I2	Male	19–21	13–15
I3	Male	19–21	10–12
I4	Female	19–21	10–12
I5	Female	19–21	13–15
I6	Female	19–21	10–12
Mean		20.8 (SD 0.6)	12.5 (SD 0.9)

**Table 2 sports-12-00038-t002:** Examples of questions used in this study.

Topic	Example Questions
Background	At what age did you start with biathlon? Were you engaged in other sports at the same time? If not, at what age did you specialize exclusively in biathlon?
Shooting training	What do you consider to be the most crucial aspects of shooting training? Can you describe what a typical shooting exercise looks like for you? Reflecting on your current knowledge, are there any aspects of your junior year shooting training that you would have approached differently? What has been the most important type of training to develop your shooting skills during your junior years?
Development	What examples can you give of structuring a shooting session (content, execution, assistance from others, etc.) to best develop your core shooting skills? In hindsight, is there anything you regret not prioritizing more or less in your shooting training during your junior years?
Cross country part vs. shooting	What do you consider your strength as a biathlete? Which of these two (cross country vs. shooting) is more important? Why? Have your perceptions about this changed during your career as an athlete?
Mental part	What do you consider the most crucial aspects of developing shooting skills? Additionally, could you highlight the critical elements of your approach to shooting training? (probing also about mental skills for both questions)
Coaching and training environment	How do you prefer to receive assistance from your coach during a shooting session? Why? How do your coach(es) balance training between cross-country skiing and shooting? Do you feel this approach works effectively? How can a coach best organize shooting training for 6–10 individuals? Moreover, 20 athletes—what would be different?

## Data Availability

The data are not accessible to the public due to the absence of data sharing permissions in the initial study ethics submission and participant consent form.

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
