# Peer review of "Psychological Determinants in Biathlon Performance: A U23 National Team Case Study"

_sports, 2024, doi:10.3390/sports12020038_

Round 1

Reviewer 1 Report (Previous Reviewer 3)

Comments and Suggestions for Authors

Dear Authors,

Your article quality has definitely improved. The introductory part prepared much clear. The Self-Efficacy theory is presented good, the importance of its application in sports is explained

Regardless, the methodology part is still lacking the presentation of the validity/trustworthiness of the study.

I would like to make a remark about the literature source in Lithuanian.  I would recommend presenting it correctly. I offer the following presentation: 

Kočergina, N. (2015). Skirtingo meistriškumo biatlonininkių (-ų) sportinio rengimo modeliavimas: disertacija [Athletic preparation modeling of female and male biathletes of different performance levels: Dissertation]. Kaunas: LSU

Taking these comments into account, the article will be prepared for publication.

Author Response

Dear Reviewer 1

We sincerely appreciate your considerate review and valuable feedback throughout the review process. Your insights have significantly contributed to improving the overall quality of the article and we have made the enhancements in the latest review as asked for. In the methodology section, we did our best to address the validity and trustworthiness by including a paragraph outlining this. We hope this addition bolsters the methodological foundation of our research. We also appreciate your recommendation for the citation in Lithuanian literature and revised it as recommended.

We have also made changes in the reference list to follow the MDPI Sports guidelines and included the tables within the manuscript body.

We hope these revisions align with the expectations, thus enhancing the scholarly merit of the manuscript.

Thank you once again for your thoughtful evaluation. We look forward to any additional feedback you may have.

Reviewer 2 Report (Previous Reviewer 1)

Comments and Suggestions for Authors

I have the following comments on the text:

Before commencing our research, we followed the requirements of the Norwegian Centre for Research Data (NSD, now Sikt.no) and the University and collected the necessary permissions.

What approvals did the authors gather? It should be clearly written whether there was approval from the BIoetical Commission? If so, add the consent number. If not, add information on which permissions and which numbers were given for the permissions.

Bibiography does not meet the style of the journal and the tables also do not meet the graphic style of the journal.

Best regards

Author Response

Dear Reviewer 2

Thank you for your valuable feedback on our manuscript throughout the review process.

For our study, we followed the requirements of the Norwegian Centre for Research Data (Sikt.no), which is the designated authority for approving social science research. Being a research work in social sciences in Norway, we did not (and could not) seek approvals from the University or a Biomedical Committee, as our research falls within the scope of Sikt.no's jurisdiction. If the study had involved health-related or particularly sensitive data, the necessity for Biomedical Committee approval would have been acknowledged. However, we have included the reference number for this approval - which we apologize that we forgot in the previous manuscript version. It should be included as asked for, so again thank you.

Additionally, we have thoroughly revised the bibliography to align with the journal's style and ensured that the tables now adhere to the journal's requirements (included in the text body). We believe these changes enhance the clarity and conformity of the manuscript to the journal's standards. For better readability, we did not show the track changes in the reference list, but we hope all changes make the manuscript within the MDPI Sports expectations.

Again, thank you for the thoughtful evaluation. We look forward to any additional feedback you may have.

This manuscript is a resubmission of an earlier submission. The following is a list of the peer review reports and author responses from that submission.

Round 1

Reviewer 1 Report

Comments and Suggestions for Authors

I have the following comments

L12 -13 – Add information about the research methods used.

L19 - add the words ''sports'' and ''biathlon''

L27-29 - add a reference of whom you are quoting.

L41-50 - add quotes.

L61 – ‘’ says, 'Here's’’ should be ‘’ says, '’Here's’’

L76 - ''e.g., '' it is not needed.

L85 - ''e.g., '' it is not needed.

Introduction - I suggest adding how many people in general practice Biathlon and what the injury rates are. I think this will improve interest in the work.

L114 – ‘’ case study’’ - add this information in the title.

L115-117 –‘’ Before commencing our research, we followed the requirements of the Norwegian Centre for Research Data (NSD, now Sikt.no) and the University and collected the necessary permissions.’’ What permissions were obtained by the authors. Describe in detail. First, whether there was approval from the local bioethics committee - add the number. Add clear information that the subjects were aware of the study and signed informed consent.

125-126 - Add tables with assigned age, gender, achievements of participants. As a designation of participants to keep them anonymous, add letter designations.

L145 - table title is missing.

References do not comply with the requirements of the journal. They should be corrected.

Reviewer 2 Report

Comments and Suggestions for Authors

Thank you for your invitation to review this article; however, I am afraid that it is not suitable for publication at this time. The most important issue is the small sample size, which does not allow valid conclusions to be reached. Please see the specific comments below:

Title

The title was too generic and not informative.

Abstract

Considering the objective, as mentioned before, the number of participants does not allow us to reach conclusions that are valid for the role population of athletes. May your purpose would be more adequate to study a limited group or representative of a population.

This conclusion does not seem adequate. Considering that the study analyzed some aspects of six people, it is not possible to draw conclusions about the implementation of a holistic approach. At most, one can conclude that one’s group has certain characteristics.

Introduction

Considering that biathlon is not a very popular sport, I suggest to present it and provide some basic information.

Reviewer 3 Report

Comments and Suggestions for Authors

Dear authors,

The topic presented in the article is a relevant topic not only in specific sports as in  biathlon, but also in all sports.

The study has much potential, notwithstanding, text corrections in study parts Introduction, Materials and Methods are necessary.

The introductory part lacks a deeper analysis of the influence of psychological factors on sports results, reviews of already conducted research. The Self-Efficacy theory is presented quite superficially. The possibilities and importance of its application in sports are unclear.

In Material and Methods part the philosophical position of research is missing. The authors not explaining the ontological and epistemological assumptions and how they guided the research. It is not entirely clear what method the authors used to analyze the data. This part misses the presentation of the validity/trustworthiness of the study.

Authors should review the reference list more carefully. (references no. 1, 2). Source 18 is cited incorrectly. There are also doubts as to whether the authors can understand a scientific work written exclusively in Lithuanian.

I hope these insights will allow the authors to improve the article